# Engineering electro-crystallization orientation and surface activation in wide-temperature zinc ion supercapacitors

Lulu Yao [1], Nandu Koripally [2], Chanho Shin [1], Anthony Mu [3], Zheng Chen [1,3,4], Kaiping Wang[1] & Tse Nga Ng [1,2,4] ✉

Matching the capacity of the anode and cathode is essential for maximizing electrochemical cell performance. This study presents two strategies to balance the electrode utilization in zinc ion supercapacitors, by decreasing dendritic loss in the zinc anode while increasing the capacity of the activated carbon cathode. The anode current collector was modified with copper nanoparticles to direct zinc plating orientation and minimize dendrite formation, improving the Coulombic efficiency and cycle life. The cathode was activated by an electrolyte reaction to increase its porosity and gravimetric capacity. The full cell delivered a specific energy of $192 \pm 0.56$ Wh kg$^{-1}$ at a specific power of 1.4 kW kg$^{-1}$, maintaining 84% capacity after 50,000 full charge-discharge cycles up to 2 V. With a cumulative capacity of 19.8 Ah cm$^{-2}$ surpassing zinc ion batteries, this device design is particularly promising for high-endurance applications, including un-interruptible power supplies and energy-harvesting systems that demand frequent cycling.

Rechargeable electrochemical cells with zinc metal anodes[1–3] are appealing due to their low cost, safety, and high divalent capacity, but many demonstrations used a thick zinc foil to compensate for the depletion of active zinc upon redox cycling. The surplus of zinc led to a large imbalance in the capacity ratio of negative to positive electrodes (n/p), thereby diminishing the overall specific energy in a full cell. As shown in Fig. 1, a device with an ideal n/p ratio of 1 would theoretically achieve a cell capacity up to three times greater than a device with a n/p ratio of 10. Therefore, addressing the challenges to realize cells with a more balanced n/p ratio is critical for their practical implementation. This study presents two strategies to correct the n/p imbalance, by decreasing the need for excess zinc in the anode and increasing the capacity of activated carbon in the cathode.

For the negative electrode, one approach to reduce zinc is through the "initially anode-free" design[4–7], in which the electrode starts out with a current collector only, excluding redox-active or host materials. During the initialization step, metal ions from the electrolyte or cathode are deposited onto the current collector, forming a thin metal layer that serves as the active material for subsequent redox cycles. This design eliminates the use of thick metal foils and offers straightforward adjustment of anode capacity through tuning the deposition time. However, zinc anodes prepared in this manner are susceptible to dendrite formation which diminishes the limited supply of active zinc and quickly causes short circuits and device failure. To mitigate dendrites, the key is to encourage planar, compact grain growth of the Zn(002) facet[8–13]. While previous works have controlled the electro-crystallization orientation by using three-dimensional architectures[14,15] or epilayers[16–18] on current collectors, they entail complex fabrication that is difficult to scale up for mass production. Here we introduce a scalable method to modify the current collectors with sputtered copper nanoparticles, pinpointing a size effect that strongly promotes Zn(002) deposition.

For the positive electrode, we chose to use low-cost activated carbon (AC), because AC supports fast kinetics and enables high charging/discharging current densities known to inhibit dendrite growth[19,20]. However, the commercially available AC shows a lower

[1]Program of Materials Science and Engineering, University of California, San Diego, La Jolla, CA, USA. [2]Department of Electrical and Computer Engineering, University of California, San Diego, La Jolla, CA, USA. [3]Aiiso Yufeng Li Family Department of Chemical and Nano Engineering, University of California, San Diego, La Jolla, CA, USA. [4]Sustainable Power and Energy Center, University of California San Diego, La Jolla, CA, USA. ✉e-mail: tnn046@ucsd.edu

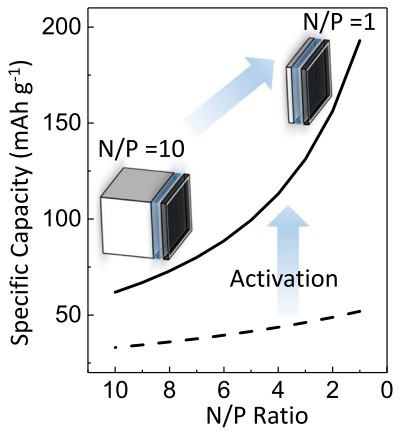
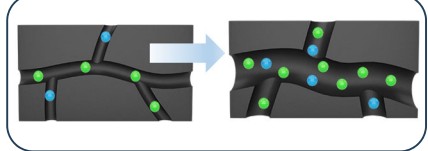
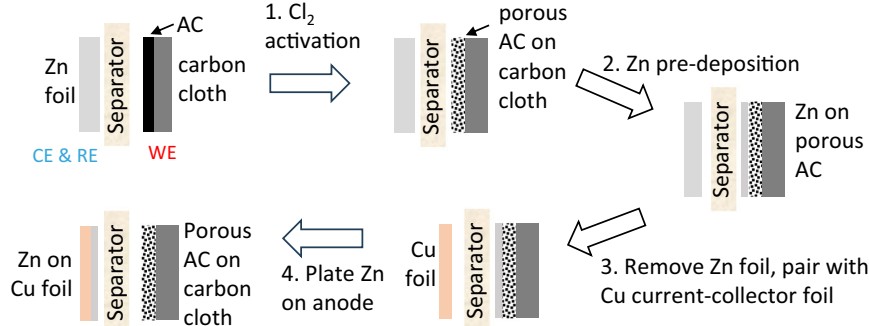

**Fig. 1 | Strategies to raise the gravimetric capacity of zinc ion supercapacitors.** It is desirable to balance the n/p ratio towards 1. To decrease the reliance on excessive zinc in the "n" part of the ratio, copper nanoparticles are deposited on the current collector to favor electro-crystallization of Zn(002) which increases Coulombic efficiency and extends cycle life. For the "p" component, the cathode capacity is enhanced by an activation method based on gas evolution that cracks additional pores in the activated carbon electrode. The calculations used for the device capacity plot are detailed in the "Methods section".

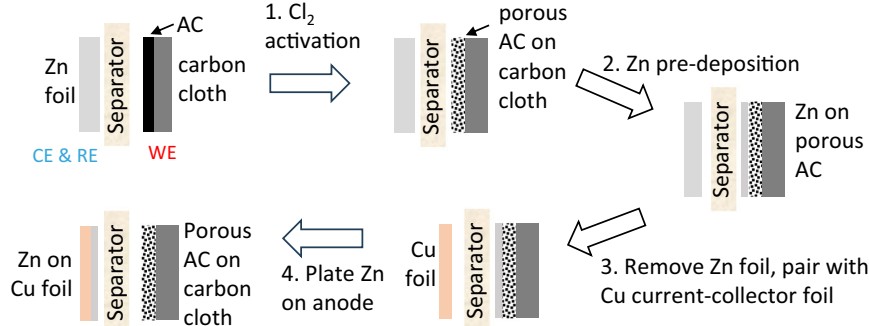

**Fig. 2 | Process flow for the fabrication of supercapacitors.** Step 1: Cl$_2$ activation of the active carbon (AC) at the cathode. Step 2: Deposit Zn onto the cathode for subsequent transfer to the anode side. Step 3: Assemble the cathode with a new separator, fresh electrolyte, and a new current collector on the anode side. Step 4: Strip Zn from the cathode, which then deposits Zn onto the anode current collector to complete the device fabrication.

gravimetric capacity than specialty carbon materials (graphene[21], carbon nanotubes[22]), oxides (ZnV$_2$O$_5$[23]), and redox polymers[24–27]. To improve the cathode capacity, we devise a process to increase the AC porosity through a gas-evolution reaction in a concentrated 15 M ZnCl$_2$ electrolyte. This treatment circumvents high-temperature oxidation using molten alkali oxidants in typical carbon activation[28], and it stems from opting for "water-in-salt" (WIS) ZnCl$_2$ as the cell electrolyte[29,30]. The improved capacity of the AC cathode will aid in balancing the n/p ratio.

With the optimized electrodes and WIS electrolyte, the Zn-AC supercapacitors in this work are shown to reach energy densities comparable to zinc ion batteries with high n/p ratios[31–33] while delivering much higher power densities. Moreover, the WIS electrolyte offers an extended potential window of 2 V compared to aqueous electrolytes and supports operation at extremely low temperatures down to −60 °C. The cumulative capacity, calculated by summing up the discharge capacities of all redox cycles until the capacity drops below 85% of its initial value, showcases the cycling stability of the device and its potential to enable high-endurance applications such as in un-interruptible power supplies and energy-harvesting systems.

## Results

### A. Process flow for the fabrication of supercapacitors

Our supercapacitor is a hybrid electrochemical cell combining different charge-storage mechanisms, which are redox reactions at the anode and electric double layers (EDL) at the cathode. We use the notation that during discharge, an anode corresponds to a negative electrode, and a cathode to a positive electrode. The device fabrication process is detailed in Figure S1, and below is a brief summary as illustrated in Fig. 2.

The preparation of the cathode involved two steps, firstly the AC activation by Cl$_2$ gas evolution and secondly the pre-deposition of zinc for future transfer onto the anode current collector. The AC activation was carried out by applying a high voltage >2.2 V to the cathode, causing electrolyte decomposition into Cl$_2$ gas that created or expanded pores in the AC and thereby increased its EDL capacity. In our configuration, the amount of zinc available for redox cycling is set through a pre-deposition process carried out on the cathode, by adjusting the deposition time and current density.

Following the above two steps, the cathode was placed in a new cell assembly, where the anode initially consisted of only a current collector. Then the cathode was set to 2 V to strip zinc off the cathode and plate it onto the anode current collector. When the zinc on the cathode was completely removed, the anode reached a fully charged state, and the device fabrication was finished.

### B. Improving Coulombic efficiency by controlling electro-crystallization orientation at the anode

This study incorporated two innovations in the fabrication process flow: one aimed at improving the Coulombic efficiency and cycling stability at the anode, and the other designed to boost the capacity at the cathode. Specifically for the anode, the surface of the copper current collector was modified with copper[34–37] nanoparticles (CuNPs)

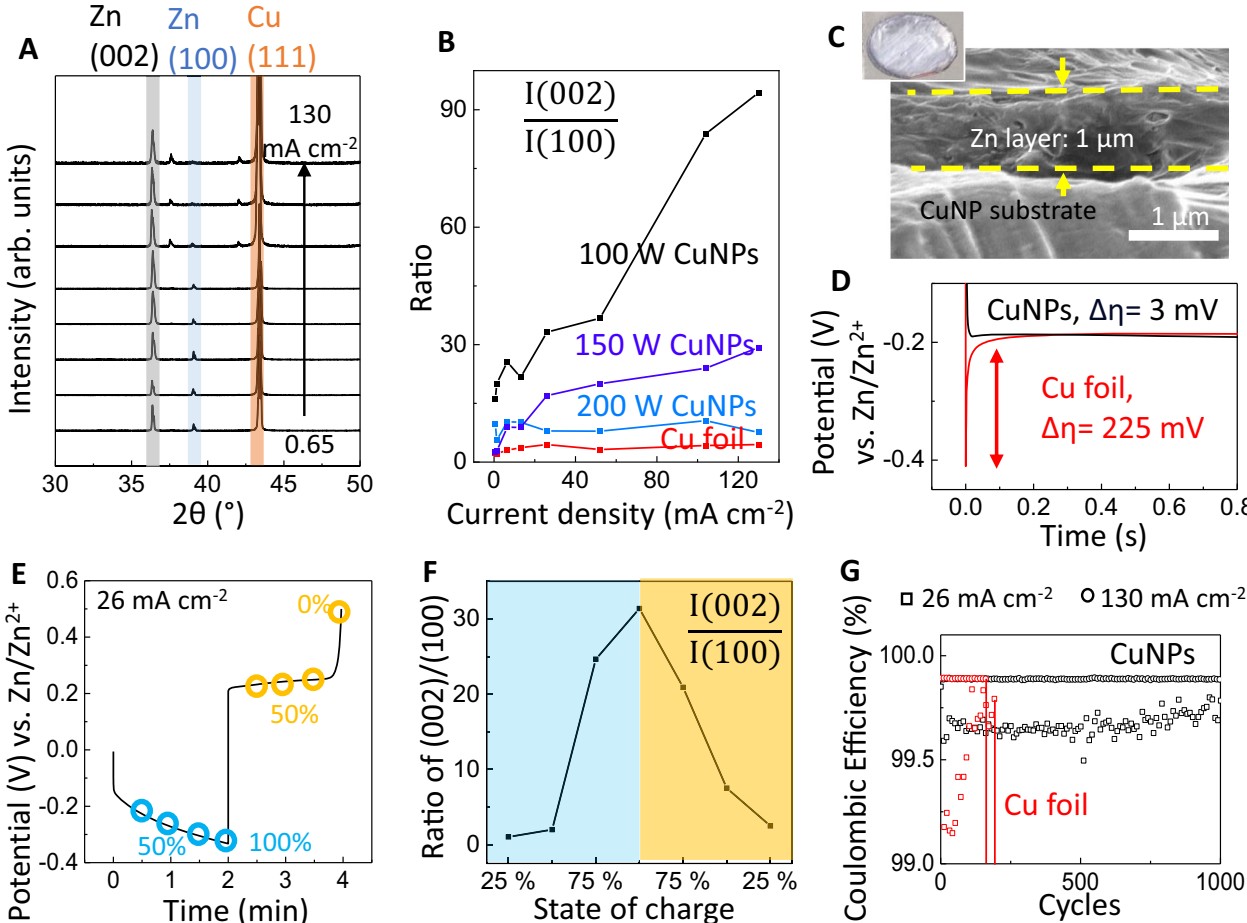

**Fig. 3 | Characterization of zinc orientations under various plating/stripping conditions. A** X-ray diffraction patterns of zinc plated at different current densities (0.65–130 mA cm$^{-2}$) on a copper current collector with CuNPs sputtered at 200 W. The zinc loading capacity was 0.85 mAh cm$^{-2}$. **B** Peak intensity ratio of Zn(002)/Zn(100), as a function of the deposition current density and CuNPs modification. **C** SEM images of the anode cross-section. Inset: a photo of the anode 1 cm in diameter. **D** Voltage during zinc plating on the bare Cu versus CuNPs-modified current collector, at a current density of 130 mA cm$^{-2}$. **E** Voltage at the 1000$^{th}$ redox cycle on the CuNPs-modified current collector, at a current density of 26 mA cm$^{-2}$. **F** Peak intensity ratio of Zn(002)/Zn(100) measured on the sample in part e. **G** Coulombic efficiency as a function of redox cycles and plating/stripping current densities, on the bare Cu (red) and CuNPs-modified (black) current collectors. At 100% state-of-charge, the anode capacity was 0.85 mAh cm$^{-2}$, measured with an electrolyte of 200 μL 15 M ZnCl$_2$.

deposited by radio-frequency (RF) sputtering. The approach of coating nanoparticles from solution onto current collectors had issues with uneven particle distribution and missing coverage. Instead of solution coating, the RF sputtering process allowed more uniform deposition, where the particle size of the CuNPs was tuned by changing the sputtering power and time. A uniform coverage extended over large areas of tens of square centimeters and was completed within a few minutes, making this process scalable for fast fabrication. The size distribution of CuNPs was easily controlled by adjusting the RF sputtering power. As shown in Figure S2, the CuNPs decreased in size with lower RF power, where the particle diameters were ~200 nm at 200 W, ~50 nm at 150 W, and <20 nm at 100 W. Below we investigated how the CuNPs guided the zinc orientation during electro-crystallization, which in turn played a crucial role on dendrite formation, Coulombic efficiency, and the cycling stability of the anode.

Figure 3A displays an example of x-ray diffraction (XRD) patterns measured on an anode after zinc was plated at various current densities. The XRD peaks corresponding to the Zn(002) and Zn(100) facets are denoted by gray and blue bands, respectively. The ratio of peak intensity $I$(002)/$I$(100) was extracted to compare the relative prevalence of these two facets depending on CuNPs surface modification and the plating current density, as shown in Fig. 3B (primary

XRD data are included in Figure S3). The $I$(002)/$I$(100) ratio was at its minimum on the bare Cu current collector and increased upon modification by sputtering CuNPs, with the smallest CuNPs deposited at the lowest RF power leading to the highest ratio. This trend indicated a strong orientation effect from the CuNPs promoting zinc electro-crystallization in the (002) plane over (100), providing a handle to encourage compact planar growth seen in Fig. 3C and reduce dendrites in (100) and (101) orientations during zinc plating. Due to the overlap of the XRD peak for Zn(101) with that of Cu(111), we replaced the Cu current collector with graphite or aluminum foil to unmask the Zn(101) peak in Figure S4. Similar to the case with Zn(100), the intensity ratio between Zn(002) and Zn(101) decreased upon modification with CuNPs, verifying that CuNPs on the surface were effective across various current-collector substrates in suppressing dendritic growth along both (100) and (101) directions.

Despite the sputtered CuNPs being randomly oriented without the well-defined crystal facets used in previous studies to mitigate dendrites, they directed a preference to (002) growth parallel to the substrate. The CuNPs provided additional nucleation sites which lowered the plating overpotential Δη, defined as the potential difference between nucleation and growth processes[8,38]. In Fig. 3D, the Δη for zinc deposition on an anode with CuNPs was just 3 mV, significantly less

than the 225 mV observed on bare Cu. This suggested that it was energetically favorable for zinc to nucleate and grow laterally on CuNPs across the surface, resulting in (002) horizontal platelets. The decrease in overpotential was similarly observed in ref. 39, which was correlated to preferential Zn(002) deposition over Cu(100) compared to a routine Cu foil. In contrast, the substantial potential barrier for nucleation on bare Cu led to vertical growth and a low $I(002)/I(100)$ ratio as seen in Fig. 3B. Furthermore, reducing the CuNPs size decreased the $\Delta\eta$, evident in the voltage measurements and scanning electron microscopy (SEM) images in Figures S5 and S6, respectively, contributing to larger (002) grains in the plated zinc.

Complementary to this energy view, another work[40] presented finite element analysis (FEA) simulations that showed a more uniform electric field distribution due to nanowire or 3D scaffolds. This electric field effect might be applicable to CuNPs, since they would likely change the surface electric field in a similar manner. Another consideration is the binding energy of Zn on difference Cu surfaces. From density functional theory (DFT) calculations, the edges of Cu nanowires would reduce Zn binding energy[40]. A similar mechanism might apply to CuNPs due to the increase in edges, promoting zincophilicity and dense lateral plating.

In addition to the effect from CuNPs size, the electrocrystallization of (002) facets also increased with the plating current density. The use of high current had been reported[19,41] to reduce dendrites and promote the formation of a dense zinc layer by offsetting the growth rate differences among various orientations. Here the application of a high current density accelerated deposition and further enhanced the (002) preference induced by the CuNPs, elevating the $I(002)/I(100)$ ratio from 15 to 94 as the current density was raised from 0.65 mA cm$^{-2}$ to 130 mA cm$^{-2}$ in Fig. 3B.

Steering zinc plating towards the (002) orientation conferred several advantages, including a dendrite-free morphology leading to better reversibility of the plating/stripping cycles and increased zinc utilization. For example, the reversible electrical and morphological characteristics are presented in Fig. 3E, F, respectively, measured after 1000 plating/stripping cycles at a current density of 26 mA cm$^{-2}$. At the beginning of zinc deposition with the anode at a state-of-charge (SOC) of 25%, the $I(002)/I(100)$ ratio was 1.03. Then the ratio increased to 31.4 by the time the SOC reached 100% (primary XRD data in Figure S7). During the stripping process, the $I(200)/I(100)$ returned to 1 when the SOC decreased to 0%. There could be alloy formation between CuNPs and Zn, which may affect the nanoparticle size, but from the SEM in Figure S2, the CuNPs looked intact after 1000 redox cycles. These data confirm that preferential (002) deposition persisted even after an extensive 1000 cycles, conducted at 100% utilization of the deposited zinc (Figure S8 shows the cut-off potential set to 0.7 V vs Zn/Zn$^{2+}$ for a full discharge).

In Fig. 3G, the Coulombic efficiency (C.E.) was 99.6% (squares, 26 mA cm$^{-2}$) to 99.9% (circles, 130 mA cm$^{-2}$) for the anode with CuNPs after 1000 cycles, while the anode with a bare Cu surface short-circuited after just 200 cycles, irrespective of the applied current densities being high or low. While the CuNPs and bare Cu surfaces showed similar C.E., the difference was the extended cycle life for the CuNPs surface due to dendrite suppression Thus, the modification with CuNPs on the anode current collector has effectively addressed the dendrite issue and significantly improved cycling stability, to open up the opportunity to use less excess zinc and achieve a more balanced n/p ratio in a full cell.

## C. Increasing gravimetric capacity by Cl$_2$ gas evolution at the cathode

For the activated carbon on cathode[42], we processed the as-purchased AC by an additional activation step that involved deliberately decomposing the 15 M ZnCl$_2$ electrolyte, which would be discarded afterwards (a fresh electrolyte would be used for the full-cell fabrication). In Fig. 4A, when the applied voltage on the cathode exceeded 2 V, a pronounced oxidation peak appeared, corresponding to the oxidation of Cl$^-$ ion into Cl$_2$ gas that would expand the porosity in the AC electrode. After repeated cycling between 1 V and 2.5 V for 1000 times at a constant current of 26 mA cm$^{-2}$, the AC morphology showed a more porous surface in the SEM Fig. 4B, C. There was a rapid increase in the capacity of AC during the first two hours of Cl$_2$ activation process, and then the process reached a plateau by the third hour as shown in Figure S9.

The electrode porosity was estimated from Brunauer-Emmett-Teller (BET) analysis shown in Figure S10 and Table S1. The active surface area of the AC on a carbon cloth substrate was measured to be 1670 m$^2$ g$^{-1}$ for the as-purchased AC, in good agreement with the specification of 1600 m$^2$/g provided by the vendor Kuraray. Then the surface area of this sample increased by 3.4 times to 5750 m$^2$ g$^{-1}$ after the Cl$_2$ activation. The pore radius also expanded to >1 nm after Cl$_2$ activation, confirming the larger pores seen in the SEM images.

Following the Cl$_2$ activation process, cyclic voltammetry (CV) in Fig. 4D showed that the electrode capacitance increased from originally 114 F ± 0.15 g$^{-1}$ up to 417 F ± 2.7 g$^{-1}$, likely due to the enhanced porosity conducive to penetration and transport of electrolyte ions in the AC. With the galvanostatic charge-discharge (GCD) measurements, Fig. 4E shows the gravimetric capacity of the cathode as a function of the applied current density (raw GCD in Figure S11). After Cl$_2$ activation, specific capacity of the carbon-cloth current-collector is 0.147 mAh g$^{-1}$, negligible compared to the specific capacity of AC. At a low current density of 1 mA cm$^{-2}$, which was equivalent to 0.8 A g$^{-1}$, the cathode capacity was 256 ± 2.7 mAh g$^{-1}$, an improvement over another treatment method that yielded rubidium-activated porous carbon[43] with 216 mAh g$^{-1}$. When the applied current density was increased by a 100 fold (to 102 mA cm$^{-2}$ or 80 A g$^{-1}$), the cathode capacity still reached 84 mAh g$^{-1}$, allowing the AC to deliver high energy even at large power output. The cathode also maintained 100% of its capacity after 10,000 charge-discharging cycles between 0 and 2 V (Figure S11), so its stability was unaffected by the Cl$_2$ activation step. The standard electrode potential of reaction 2Cl$^-$ − 2e$^-$ → Cl$_2$ is 1.396 V vs. SHE (2.16 V vs. Zn/Zn$^{2+}$), higher than our 2 V device operating window to avoid potential Cl$_2$ evolution during operation[44,45].

## D. Performance metrics of Zn-AC supercapacitors

The supercapacitors were prepared at a n/p ratio of 4 and 2.5 (n: Zn; p: AC), to demonstrate high utilization of active materials. The n/p ratio was calculated based on the areal capacities of the anode and the cathode measured by potential versus time characteristics, detailed in Figure S12. In Fig. 5A, as the n/p ratio was adjusted from 4 to 2.5, the areal capacity of the full cell increased by 2.87 times from 0.15 mAh cm$^{-2}$ to 0.43 mAh cm$^{-2}$, while the equivalent gravimetric capacity was raised from 82.5 mAh g$^{-1}$ to 98.3 mAh g$^{-1}$. These measured gravimetric capacities were close to 70% of the theoretical prediction in Fig. 1.

Lowering the n/p ratio was desirable for improving the device capacity, although it conversely affected the cycling stability. Following 50,000 full charge-discharge cycles (each discharge was ~55 s at 26 mA cm$^{-2}$, from 2 V to 0 V), equivalent to 763 h of discharge, the capacity retention was 92% for n/p = 4 and 84% for n/p = 2.5, on account of less excess zinc available to make up for losses caused by side reactions and dendrite disconnections. Therefore, we chose not to decrease the n/p ratio further below 2.5. Nonetheless, those high retention rates of >80% after 50,000 cycles demonstrated stability already, attributed to the CuNPs modification of the current collector to suppress zinc dendrites. By comparison, the device with an unmodified current collector dropped to 35% of its initial capacity after 10,000 cycles (Figure S13). Furthermore, the stabilizing effect of the CuNPs was observed in devices using different electrolytes, including aqueous zinc sulfate and organic zinc bis(trifluoromethylsulfonyl) imide (Figure S14), in addition to WIS ZnCl$_2$, to show the broad

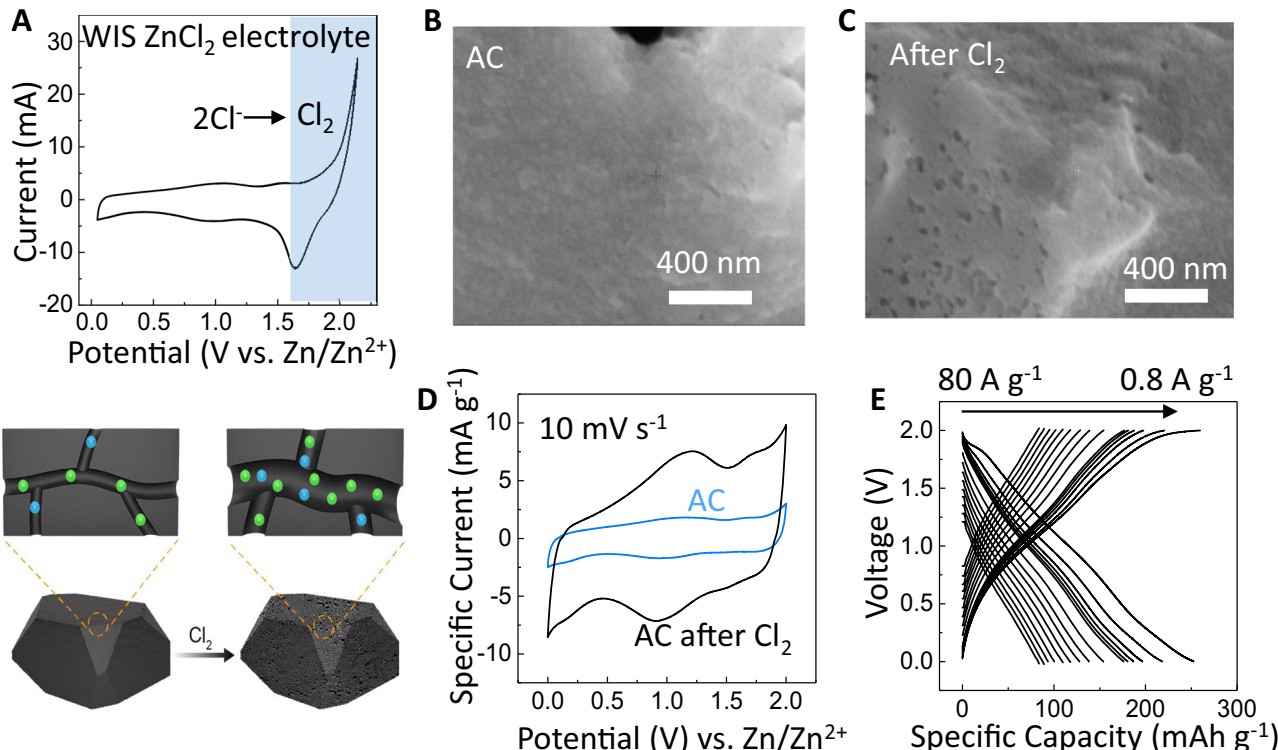

**Fig. 4 | Characterization of cathode activation. A** Cyclic voltammetry of 1.64 mg cm$^{-2}$ active carbon on carbon cloth, measured in a Swagelok cell with a 100 μm Zn foil as the anode, and 200 μL of 15 M ZnCl$_2$ as the electrolyte. When the applied potential exceeded 2 V, decomposition side reactions of the electrolyte generated Cl$_2$ gas that increased porosity in the active carbon, as drawn in the bottom illustration. Scanning electron microscopy of active carbon **B** before **C** after the Cl$_2$ activation process. **D** Cyclic voltammetry before and after the Cl$_2$ gas activation, at a scan rate of 10 mV s$^{-1}$. **E** Capacity-voltage profiles of the Cl$_2$-activated cathode at various charge/discharge current ranging from 0.8 A g$^{-1}$ to 80 A g$^{-1}$.

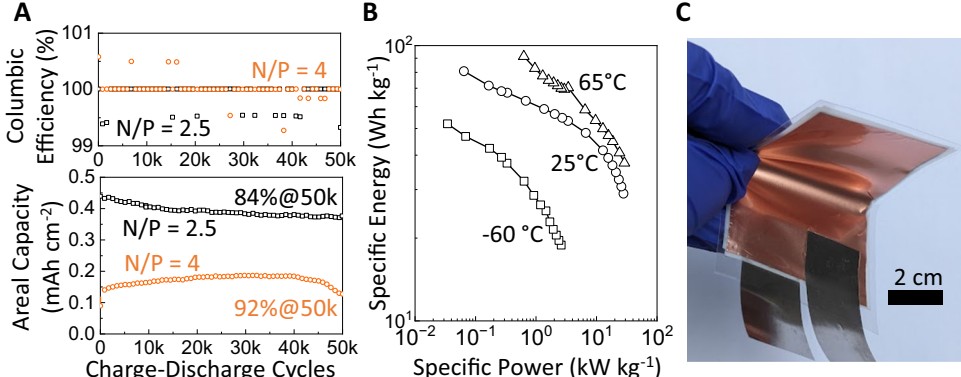

**Fig. 5 | Supercapacitor performance. A** Coulombic efficiency and areal capacity versus charge-discharge cycles, as a function of n/p ratios. The retention percentages are calculated after 50,000 cycles. **B** Specific energy versus specific power of our supercapacitor (n/p = 2.5), over a wide range of temperatures. **C** Photograph of a large 25.8 F supercapacitor (dimension: 4 cm × 5.5 cm). For part A, device with n/p = 4: 1.12 mg cm$^{-2}$ active carbon on carbon cloth as cathode; Cu current collector with CuNPs sputtered at 100 W as anode current collector; 1.41 mg cm$^{-2}$ active Zn amount; and 200 μL 15 M ZnCl$_2$ as electrolyte. Device with n/p = 2.5: 3.26 mg cm$^{-2}$ active carbon on carbon cloth as cathode; Cu current collector with CuNPs sputtered at 100 W as anode current collector; 2.54 mg cm$^{-2}$ active Zn amount; and 200 μL 15 M ZnCl$_2$ as electrolyte. For part B, device with n/p = 2.5: 1.25 mg cm$^{-2}$ active carbon on carbon cloth as cathode; Cu current collector with CuNPs sputtered at 100 W as anode current collector; 1.59 mg cm$^{-2}$ active Zn amount; and 200 μL 7.5 M ZnCl$_2$ as electrolyte. For part C, device with n/p = 2.5: 3.5 mg cm$^{-2}$ active carbon on carbon cloth as cathode; Cu current collector with CuNPs sputtered at 100 W as anode current collector; 2.69 mg cm$^{-2}$ active Zn amount; and 200 μL 15 M ZnCl$_2$ as electrolyte.

applicability of this approach in facilitating reversible zinc plating/stripping and extending the cycle life.

With its long cycle life, our device with n/p = 2.5 achieved a cumulative capacity of 19.8 Ah cm$^{-2}$, well-placed among state-of-the-art zinc ion batteries and supercapacitors in Figure S15a (also listed in Table S2; note that many of those works used only weights of the cathode for energy/power calculations). Per an individual charge-discharge cycle, the specific capacity of zinc ion batteries was about two times higher than that of zinc ion capacitors (Figure S15b). However, the long cycle life of our supercapacitor compensated for the difference, enabling it to reach a greater cumulative capacity than batteries before the device capacity fell below 85%. This feature would

be particularly beneficial for applications where the energy storage devices undergo frequent cycling, such as in energy-harvesting systems that require intermittent charging/discharging, thereby reducing the costs associated with replacing degraded devices.

Another advantage of our supercapacitor was the capability to operate across a wide temperature range of −60 °C and 65 °C, based on the use of WIS electrolyte[46]. A high concentration of $ZnCl_2$, in the WIS electrolyte minimized hydrogen bonding, allowing it to stay liquid and maintain sufficient ionic conductivity at low temperatures. To determine the optimal $ZnCl_2$ concentration, it was adjusted across a range from 1 M to 15 M. The ionic conductivity reached a peak at 7.5 M, achieving 0.1 $Scm^{-1}$ (Figure S16), but then declined with higher concentrations due to increasing viscosity with high salt content. The 7.5 M WIS electrolyte afforded a 2 V potential window and enabled the supercapacitor to operate under an extreme temperature of -60 °C with a low 8.5 Ω internal resistance (Figure S17), reaching a specific energy of 118 Wh $kg^{-1}$ at an output power of 0.08 kW $kg^{-1}$ in Fig. 5B. The charge/discharge profiles of the supercapacitor were included in Figure S17. The performance was enhanced with rising temperature, as typical for electrochemical cells[47]; at 65 °C, the specific energy rose to 208 Wh $kg^{-1}$ at 1.43 kW $kg^{-1}$, demonstrating wide temperature tolerance.

Since our processing methods are scalable to large area, we fabricated a large device with an electrode area of 4 cm × 5.5 cm in Fig. 5C. The device, with a n/p ratio of 2.5, was measured to have a capacitance of 25.8 8 ± 0.2 F and a capacity of 14.4 ± 0.11 mAh. This capacity is practical for commercialization as it is comparable to a Maxwell supercapacitor product (model# BCAP0025 P270 S01) in capacitance, while our device showed a higher volumetric capacity by two fold. Notably, after 1000 charge-discharge cycles, the device retained 95.8% of its initial capacity (Figure S18), indicating outstanding cycling stability and uniform zinc plating/striping over a substantial area. The scalability of our device is promising for future manufacturing.

In Figure S15c, the energy and power densities of our large supercapacitor are superior to previously reported zinc ion supercapacitors (data in dark blue color). Compared to batteries (light blue data) at low power output, our device showed a lower specific energy; yet at a higher power output of 1.4 kW $kg^{-1}$, our device was better with a specific energy of 192 ± 0.56 Wh $kg^{-1}$, surpassing the $K_{0.486}V_2O_5$||Zn battery with 109 Wh $kg^{-1}$ [33]. Our supercapacitor design offers high-power/current outputs that batteries cannot match, and the high current/power densities also promote dense Zn(002) deposition, which helps to prevent zinc dendrite formation and extend the lifetime.

In summary, this work presents design guidelines for zinc ion supercapacitors, where the current collector was modified with CuNPs to guide the orientation of zinc plating, alongside a $Cl_2$ activation method developed to improve the gravimetric capacity. The resulting devices achieved a low n/p ratio of 2.5 with long cycle life. The supercapacitors provided high specific power and fast kinetics to meet peak power demands. Though the specific energy per charge cycle was lower, the cumulatative capacity of the supercapacitors was superior to typical zinc ion batteries and would reduce replacement costs. Thus, the devices in this study are particularly promising for applications that involve daily cycling in energy-harvesting systems, or in remote infrastructures where maintenance access would be difficult.

## Methods

### Current collector preparation
Commercial Cu foils (MTI, 10 μm thick, 99.99%) were used as the current collector in the initially anode-free devices. The surface modification with CuNPs was prepared by RF sputtering on the Cu foil with the Denton Discovery 18 sputter system.

### Preparation of AC electrodes
The electrode slurry was prepared by grinding AC (YP50F, 1600 $m^2 g^{-1}$, Kuraray), PVDF (MTI, Lib-PVDF), and carbon black (MTI, Lib-SP)

powders together in a mortar and pestle at the weight ratio of 7:2:1. The powder slurry was mixed with n-methylpyrrolidone (Sigma, 99.5%) as the solvent. The slurry was stirred in a parafilm-sealed beaker at 100 rpm for 12 h in ambient conditions. The slurry was coated on the surface of carbon cloth (AvCarb, 1071HCB) by screen printing with a doctor blade. Carbon cloth was selected here due to its high electrochemical stability when the potential is > 1 V vs. Ag/AgCl. After drying at 80 °C in ambient air for 12 h, the electrodes were ready to be used for experiments.

### Activation of AC electrode
In the activation process, in a two-electrode configuration, an AC electrode was used as the cathode, a zinc foil (ThermoFisher, 011912-HG, 99.99%) was used as the counter and reference electrode, and 15 M $ZnCl_2$ was used as the electrolyte. The assembled cell was charged/discharged between 1 V and 2.5 V vs. $Zn/Zn^{2+}$ at a constant current density of 26 mA $cm^{-2}$. The charging/discharging process continued until the columbic efficiency of device stabilized over time.

### Device fabrication
As shown in Fig. 2 and Figure S1, the device initially consisted of $Cl_2$ activated AC as the cathode, with zinc to be plated onto the CuNPs modified current collector as the anode, a glass microfiber filter (Sigma, Whatman GF/A) as the separator, and 15 M $ZnCl_2$ (TCI, 98%) as the electrolyte, at a volume of 200 μL. Small devices were assembled and tested inside a Swagelok cell. Two glass carbon electrodes (Nanochemazone, 7440-44-0, 99%) were used as contacts to the electrodes. For the large device in Fig. 5D, Titanium foil (ThermoFisher, 13 μm thick, 044243-6 N, 99.6%) was used as the current collector for both cathode and anode side. The encapsulation was conducted by heat lamination of polyethylene terephthalate (PET) film at 110 °C.

### Gravimetric capacity equation for an anode-free device as a function of the n/p ratio and electrode capacities for Fig. 1
The following calculations assign 1 gram as the weight of the cathode. The measured capacity of active carbon before activation is 55.42 mAh $g^{-1}$, and after activation is 252.39 mAh $g^{-1}$. The theoretical capacity of Zn anode is 820 mAh $g^{-1}$. Subscript $c$ stands for the cathode, $a$ for the anode, $dev$ for the full device, and $g$ for gravimetric, n/p for n-to-p capacity ratio. $Cap$ is capacity and $m$ is mass.

$$Cap._{Dev,g} = \frac{Cap._{C,g}}{m_{Dev}} \tag{1}$$

$$m_{Dev} = m_C + m_A \tag{2}$$

$$Cap._A = Cap._{C,g} \times 1\,gram \times \frac{n}{p} \tag{3}$$

$$m_A = \frac{Cap._A}{Cap._{A,g}} = \frac{Cap._{C,g} \times \frac{n}{p}}{Cap._{A,g}} \tag{4}$$

$$m_{Dev} = m_C + m_A = 1\,gram + \frac{Cap._{C,g} \times n/p}{Cap._{A,g}} = \frac{Cap._{C,g} \times \frac{n}{p} + Cap._{A,g}}{Cap._{A,g}} \tag{5}$$

$$Cap._{Dev,g} = \frac{Cap._{C,g}}{m_{Dev}} = \frac{Cap._{C,g} \times Cap._{A,g}}{\frac{n}{p} \times Cap._{C,g} + Cap._{A,g}} \tag{6}$$

## Materials characterization

FEI scanning electron microscope at 5 kV was used for capturing materials morphology. Anton Paar XRDynamic 500 was used to collected the XRD data from three different locations for each sample.

Anton Paar™ Quantachrome autosorbIQ with nitrogen gas was used to collect the BET data for carbon cloth and AC samples before and after $Cl_2$ activation. The samples were degassed at 200 °C for 20 h and reweighted to determine the sample mass. Nitrogen adsorption was performed at 77 K. The specific surface area was calculated from sorption isotherms using BETSI program and Rouquerol criteria.

## Electrochemical characterization

All half-cells and full cells were fabricated and tested under room atmosphere. The characterization of cathodes, anodes, and full cells was performed on more than three samples. Each electrochemical characterization method (CV, GCD, EIS, etc.) was repeated at least three times per sample, and the reported values were the average. The extracted standard deviations were used for the ± error values. Long-term stability tests (50k cycling) were conducted on two devices with different loadings, constrained by the limited resources available for extended testing (hundreds of hours for each measurement).

Characterization of anodes: Electrochemical measurements were carried out via a BioLogic SP-200 potentiostat. The characterization of anodes was conducted in Swagelok cell in two-electrode configuration with a 100 μm zinc foil as the counter and reference electrode, and 15 M $ZnCl_2$ as the electrolyte. Constant current deposition process was conducted to deposit zinc on the surface of current collectors. To determine the Columbic efficiency of the electrode, zinc was deposited to the capacity of 0.45 mAh cm$^{-2}$ on the current collector under different current densities, and then it was stripped away to a cutoff potential of 0.6 V vs. $Zn/Zn^{2+}$ at the same current density.

The Coulombic efficiency was measured with a half-cell setup with Zn foils serving as the counter and the reference electrodes. And, the Coulombic efficiency of the device = (stripping capacity)/(deposition capacity), with the capacities calculated from the GCD characteristics.

Characterization of cathodes: Cathodes were tested in the two-electrode configuration in a Swagelok cell, with AC on carbon cloth (loading varied in the range of $1.26 - 4$ mg cm$^{-2}$) as the working electrode, a 100 μm zinc foil as the counter and reference electrode, and 15 M $ZnCl_2$ as the electrolyte.

Characterization of supercapacitors: The n/p ratio was first calculated by measuring the capacities of anode and cathode separately at a current density of 0.65 mA cm$^{-2}$. The ionic conductivities of $ZnCl_2$ electrolytes at different concentrations and temperatures were inferred from electrochemical impedance spectroscopy performed at 0 V with an amplitude of 10 mV and at frequencies ranging from 1 MHz to 100 mHz. The equivalent series resistance of the electrolyte samples was used to calculate ionic conductivity.

Characterization of ionic conductivity and supercapacitor at different temperatures: The impedance and GCD measurements from -60 °C to 65 °C were conducted with a commercial freezer and a static heating oven. The temperature of the freezer was controlled by varying the amount of dry ice as monitored by a thermometer, and the oven was sealed and set to the corresponding temperature. The impedance was measured with the electrolyte and separator sealed inside a stainless-steel coin cell, and the supercapacitor was measured in the two-electrode setup in a Swagelok cell with glassy carbon electrodes. The devices were held at the chosen temperature for 20 min before starting the measurement.

For the GCD measurements, the supercapacitor was initially designed with a high n/p ratio of 4 at room temperature, to ensure that the amount of Zn was sufficient to meet the capacity of the cathode, because the cathode capacity would increase with higher temperatures. At 65 °C, the cell n/p ratio was improved to 2.5 due to this change in cathode properties with temperature.

The capacitance $C$ was calculated from cyclic voltammetry characteristics based on the equation[48]:

$$C = \frac{1}{\Delta V v} \int_{V_1}^{V_2} i \, dV \qquad (7)$$

Here $V_1$ and $V_2$ are the starting and ending potentials in the discharge portion of the measurement, $i$ is the current at each potential, $\Delta V = V_2 - V_1$ is the potential window, and $v$ is the voltage scan rate. The calculated capacitance was an average value across the whole potential window.

The specific capacitance, capacity, power and energy densities were calculated based on following equations using galvanostatic charge-discharge characteristics:

$$\text{Specific Capacitance} = \frac{I}{M} \int \left(\frac{1}{V(t)}\right) dt \qquad (8)$$

$$\text{Capacity} = I \times t_{d/M} \qquad (9)$$

and the relationship between capacity and specific capacitance is

$$\text{Specific Capacitance} = \text{Capacity} \times \frac{1}{t_d} \int \left(\frac{1}{V(t)}\right) dt \qquad (10)$$

$$E = \frac{I}{M} \int_0^{t_d} V \, dt \qquad (11)$$

$$P = \frac{E}{t_d} \qquad (12)$$

Here $I$ is the constant discharge current, $t_d$ is the time interval of the GCD discharge period, $V(t)$ is the potential as a function of time, $E$ is the specific energy, $V$ is the measured potential, $P$ is the specific power, $M$ is the mass of active materials on both cathode and anode. In Fig. 5B and S15c, the energy/power densities were calculated using only the cathode weight as done in prior works, but for completeness we also present the mass loading with both anode and cathode weights in the Table S3.

## Data availability

The source data generated in this study have been deposited in the Figshare database under accession code https://doi.org/10.6084/m9.figshare.28040456.

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

## Acknowledgements

This work was supported by the National Science Foundation (NSF) MCA 2120701 (T.N.N) and CNS 2312715 (T.N.N.). Part of the work was performed at the San Diego Nanotechnology Infrastructure of UCSD, which is supported by NSF ECCS–2025752.

## Author contributions

L.Y. conceived and carried out the experimental design, device fabrication, and characterization analysis. N.K. assisted in device fabrication and characterization analysis. C.S. assisted in device analysis. K.W. contributed to the experimental design. A. M. and Z. C. assisted in BET

characterization and analysis. T.N.N. conceived and supervised the project. All authors contributed to manuscript writing and editing.

## Competing interests

The authors declare no competing interests.
