## [Transparent Peer Review file · Nature Communications]

Engineering Electro-crystallization Orientation and Surface Activation in Wide-Temperature Zinc Ion Supercapacitors

Corresponding Author: Professor Tse Nga Ng

Version 0:

Reviewer comments:

Reviewer #1

(Remarks to the Author)

In this manuscript, the authors reported an anode-free design to balance the electrode utilization in zinc ion supercapacitors. The dendritic loss in the zinc anode is overcome by depositing Cu nanoparticles on the Cu current collector while the capacity of the activated carbon cathode is increased by in-situ Cl₂ etching treatment. Although the electrochemical data confirmed the efficacy of such strategies in both anode and cathode materials, some key details are still absent, especially on the cathode side. In my opinion, the anode modification, despite being effective, is similar to those in the previous literature (Nat. Commun. 2022, 13: 7922; Adv. Funct. Mater. 2022, 32, 2110829; Mater. Today Energy 34 (2023) 101284, etc.). However, the treatment of cathode materials is indeed interesting. So, my impression is that the manuscript could be suitable for publication in Nat. Commun. after proper revision. My concerns are as follows:

Anode side:

1. The authors should state the advantage of radio-frequency (RF) sputtering over other methods in the previous studies. It will help convince the readers to believe in the possibility of this method for scalable fabrication.
2. The concentrated ZnCl₂ solution could be an acidic electrolyte. Were the Cu nanoparticles well-kept during the charging/discharging processes? To verify this point, please provide the morphological characterization of Cu nanoparticles before and after cycling tests.

Cathode side:

1. During the etching of AC by Cl₂, was the carbon cloth (current collector) activated simultaneously? The authors should compare the electrochemical properties of bare carbon cloths before and after in situ Cl₂ treatment.
2. The authors only provided the SEM images of AC etched by Cl₂. Although the authors calculated the surface areas of the AC before and after Cl₂ treatment based on the capacitance, the deviation could exist if the carbon cloth was also activated by Cl₂. From the perspective of statistics, the quantification of specific surface area by BET test could be more accurate.
3. How about the influences of the etching time of Cl₂ on the porosity of AC? Please provide the related data.
4. When tested in a cut-off voltage of 0-2.0 V, was there the capacity contribution of Cl-/ClO redox couples on the cathode side of the Zn-ion capacitor? It is better to scrutinize the influence of the possible Cl₂ evolution.

In addition, certain word errors are present in the current manuscript version, i.e., in Figure 4d, the "CL₂" should be "Cl₂". Please double-check.

Reviewer #2

(Remarks to the Author)

In this work, the authors present an anode-free design to balance the electrode utilization in zinc-ion hybrid supercapacitors by introducing copper nanoparticles to improve the Coulombic efficiency and cycle life of zinc anodes and activating carbon cathode by an electrolyte reaction to increase the gravimetric capacity. The following serious issues limit the publication of this submission.

1. *Some key points of this work have been reported in the literature, such as the construction of activated carbon-based zinc-ion hybrid supercapacitors (e.g., doi.org/10.1016/j.ensm.2018.01.003; doi.org/10.1002/aenm.202202303), the optimization of the zinc deposition interface using Cu nanoparticles (e.g., doi.org/10.1002/adma.202300019; doi.org/10.1002/adma.202203835; doi.org/10.1002/ange.202212587), and the design of anode-free zinc-ion batteries (e.g.,*

doi.org/10.1038/s41467-022-35630-6; doi.org/10.1021/jacs.1c12764). The authors should not miss them since these representative reports are the basis of this work.

2. Since zinc was pre-deposited on the AC cathode and then deposited on the Cu foil as the anode current collector (e.g., Fig. 2), the system cannot be called "anode-free".

3. Zinc deposition corresponds to a smaller nucleation overpotential, which is favorable for fast zinc nucleation. Why is this "energetically favorable for zinc to ... grow laterally on CuNPs across the surface"? In other words, why does a smaller nucleation overpotential result in a lateral deposition of zinc? Moreover, the reason that CuNPs result in Zn(002) deposition and high Coulombic efficiency is not well revealed.

4. The test method of Coulombic efficiency in Fig. 3G should be described in detail. Besides, it seems that for the Coulombic efficiency points in Fig. 3G, some are 99.9% and some are only about 99.6%. How can the average Coulombic efficiency reach 99.9%?

5. Rate performance of the Cl₂-activated carbon cathode should be evaluated further at large current densities such as 5-20 A/g (Fig. 4E). The current density of 0.8 to 80 mA/g is too small, which cannot reflect the performance of the carbon cathode for supercapacitors. Besides, N₂ or CO₂ adsorption-desorption tests are needed to study the pore structure of the Cl₂-activated carbon material.

6. Since carbon fiber cloth was used as the current collector of the carbon cathode, its effect on the calculated capacity of the AC cathode should not be ignored. Especially the carbon fiber cloth could also be activated by the Cl₂ which provided a considerable charge-storage capacity, making the measured capacity of the AC cathode notably enhanced.

7. The charge/discharge profiles at different current densities of the "anode-free Zn-AC supercapacitors" should be provided to check the Coulombic efficiency and rate capability of the devices.

Version 1:

Reviewer comments:

Reviewer #1

(Remarks to the Author)

The authors have made a thorough revision to their paper, which should meet the high standard of Nat. Commun.. I think this manuscript can be accepted without further revision.

Reviewer #2

(Remarks to the Author)

After carefully checking the authors' responses, I still do not recommend the publication of the submission. (1) I pointed out similar works in the aspects of the construction of activated carbon-based zinc-ion hybrid supercapacitors, the optimization of the zinc deposition interface using nanoparticles and the design of anode-free zinc-ion batteries, but this work does not present enough novelty compared with these literature. (2) The concept of "anode-free" is not suitable for this work. Anode loadings are listed in Table 1, which are not zero. (3) The question that why the Cu nanoparticles are favorable for zinc to grow laterally to form (002) horizontal platelets is not well answered. (4) The authors directly changed the current density from 0.8-80 mA/g in the previous manuscript to 0.8-80 A/g in the current manuscript. How can the water-in-salt electrolyte (15 M ZnCl₂) and the highly porous carbon cathode realize such superior fast-rate performance?

We thank the reviewers and the editor for their great feedback. We have revised our manuscript accordingly and addressed the formatting requests. Thank you for your time and consideration. Our point-by-point response is below.

Reviewer #1 (Remarks to the Author):

In this manuscript, the authors reported an anode-free design to balance the electrode utilization in zinc ion supercapacitors. The dendritic loss in the zinc anode is overcome by depositing Cu nanoparticles on the Cu current collector while the capacity of the activated carbon cathode is increased by in-situ Cl₂ etching treatment. Although the electrochemical data confirmed the efficacy of such strategies in both anode and cathode materials, some key details are still absent, especially on the cathode side. In my opinion, the anode modification, despite being effective, is similar to those in the previous literature (Nat. Commun. 2022, 13: 7922; Adv. Funct. Mater. 2022, 32, 2110829; Mater. Today Energy 34 (2023) 101284, etc.). However, the treatment of cathode materials is indeed interesting. So, my impression is that the manuscript could be suitable for publication in Nat. Commun. after proper revision.

Reply: We have added the requested analysis details below. Regarding the previous works as mentioned by the reviewer, we have cited two of them in the original manuscript as Ref. 3 and 14, and added the paper in Adv Func Mater 2022, 2110829 as Ref. 34 in this revision. To contrast our difference and novelty on the anode design, the prior works were focused on electrodeposited Cu-Zn alloys or nano-Cu to mitigate dendrites. Here we demonstrated sputtering deposition method, which is fast (completed in a few minutes) and scalable to large areas for future manufacturing processes, as the reviewer pointed out. Furthermore, our method enabled good control over the crystallographic ratio Zn(002)/Zn(101), which was not achieved in the above references, and we clarified the dependence of this ratio on current density. This control allowed the device to operate at high current density and achieved high cumulative capacity.

My concerns are as follows: Anode side:

1. The authors should state the advantage of radio-frequency (RF) sputtering over other methods in previous studies. It will help convince the readers to believe in the possibility of this method for scalable fabrication.

Reply: Thanks for bringing up this advantage. We added the following text on p. 4 of the main text: “The approach of coating nanoparticles from solution onto current collectors had issues with uneven particle distribution and missing coverage. Instead of solution coating, the RF sputtering process allowed more uniform deposition, where the particle size of the CuNPs was tuned by changing the sputtering power and time. A uniform coverage extended over large areas of tens of square centimeters and was completed within a few minutes, making this process scalable for fast fabrication.”

2. The concentrated ZnCl₂ solution could be an acidic electrolyte. Were the Cu nanoparticles well-kept during the charging/discharging processes? To verify this point, please provide the morphological characterization of Cu nanoparticles before and after cycling tests.

Reply: We added SEM images of Cu nanoparticles after cycling, as Supplemental Fig S2 parts E and F which are shown below with a scale bar of 10 μm and 500 nm, respectively. The SEM images show that Cu nanoparticles were intact after 1000 redox cycles, in which the Zn were completely stripped off after discharging the anode to 0% state-of-charge. The reviewer's comment brought us to add a comment in the text on p.6 that "there could be alloy formation between Cu nanoparticles and the Zn, which may affect the nanoparticle size, but from the SEM the Cu nanoparticles looked intact after 1000 redox cycles." Nonetheless, the elemental analysis by energy-dispersive x-ray spectroscopy in the SEM did not show any Zn but only Cu at 0% state-of-charge.

Added to Supplemental Fig S2. (E, F) Surface morphology of the anode at 0% state-of-charge after 1000 redox cycles. Cu nanoparticles stayed intact on the anode current collector.

Cathode side:

1. During the etching of AC by Cl_2 , was the carbon cloth (current collector) activated simultaneously? The authors should compare the electrochemical properties of bare carbon cloths before and after in situ Cl_2 treatment.

Reply: The Cl_2 activation process did affect the carbon cloth (CC, also the current collector), but the capacity contribution of the activated CC to the whole cathode is negligible. We added this information to Supplemental Fig. S11 as shown below. The specific capacity of plain CC (mass: 9.7 mg) increases from 0.037 mAh g^{-1} to 0.147 mAh g^{-1} . However, compared to the capacity of activated AC (mass: 1.61 mg), the capacity contribution from CC is only 0.48%, confirming that most of electrode capacity comes from active material AC, not the current collector. To clarify the contribution of the current collector, we added to p.6 in the main text: "After activation, specific capacity of carbon cloth after activation is 0.147 mAh g^{-1} , negligible compared to the specific capacity of active material."

Added to Supplemental Figure S11. (C) Capacity-voltage profiles of the plain CC and Cl₂-activated CC (activated for 16 h), measured at a current density of 2 mA. (D) Capacity versus activation time, on samples of plain CC and CC with active carbon.

2. The authors only provided the SEM images of AC etched by Cl₂. Although the authors calculated the surface areas of the AC before and after Cl₂ treatment based on the capacitance, the deviation could exist if the carbon cloth was also activated by Cl₂. From the perspective of statistics, the quantification of specific surface area by BET test could be more accurate.

Reply: We agree with the reviewer and measured the BET of AC before and after Cl₂ activation, for the carbon cloth (CC) substrates and for samples of AC on CC, as shown in Supplemental Figure S10 below and in Supplemental Table S1. This discussion is added to p.6 of the manuscript. “Before Cl₂ activation, the surface area of AC on CC was measured to be 1670 m²/g, in good agreement with the specification value of 1600 m²/g provided by the AC vendor Kuraray. Then the surface area of this sample increased by 3.4 times to 5750 m² g⁻¹ after Cl₂ activation. The median pore radius also expanded to >1 nm after Cl₂ activation, confirming the larger pores seen in the SEM images.”

Added to Supplemental Figure S10. Brunauer-Emmett-Teller (BET) surface analysis. Before and

after Cl₂ activation of (A) a carbon cloth (CC) substrate, (B) activated carbon (AC) on CC substrate. (C, D) Histograms of pore radius distribution.

Supplemental Table S1. Sample surface area and porosity from BET measurements. STP: Standard temperature and pressure.

	Total pore volume at STP (cc/g)	Surface area (m ² /g)
Carbon cloth (CC)	Not applicable	6
CC after Cl ₂ activation	Not applicable	12
AC on CC	1.08	1670
AC on CC after Cl ₂ activation	13.86	5750

3. How about the influences of the etching time of Cl₂ on the porosity of AC? Please provide the related data.

Reply: We added this info to p.6 and Supplemental Figure S10: “There was a rapid increase in the capacity of AC during the first two hours of Cl₂ activation process, and then the process reached a plateau (Supplemental Figure S11).

Added to Supplemental Figure S9. (A) Cyclic voltammograms and (B) Capacity-voltage profiles of Cl₂-activated AC measured in 15 M ZnCl₂ electrolyte. The activation periods were 0 h, 0.5 h, 1 h, 3 h, and 6 h. (C) Capacity change as a function of activation time, measured at a current density of 2 mA.

4. When tested in a cut-off voltage of 0-2.0 V, was there the capacity contribution of Cl⁻/Cl₀ redox couples on the cathode side of the Zn-ion capacitor? It is better to scrutinize the influence of the possible Cl₂ evolution.

Reply: The standard electrode potential of reaction $2\text{Cl}^- - 2\text{e}^- \rightarrow \text{Cl}_2$ is 1.396 V vs. SHE (2.16 V vs. Zn/Zn²⁺), higher than our 2 V device operating window. In agreement with this, we have

observed the Cl₂ generation reaction at an onset voltage of around 2.12 V, as indicated by the figure below. The same voltage plateau region also appeared in the papers of Ref. 41 and 42 that utilized Cl⁻/Cl₀ redox couples for enhanced capacity. The reviewer brings up a good caution that to avoid possible Cl₂ evolution, we limited the potential window to 2 V. We added this caution to p.6.

modified Supplemental Fig S1 part B. Voltage region that triggers Cl₂ activation.

In addition, certain word errors are present in the current manuscript version, i.e., in Figure 4d, the “CL2” should be “Cl2”. Please double-check.

Reply: Thank you! We corrected the errors and proof-read the manuscript for typos.

Reviewer #2 (Remarks to the Author):

In this work, the authors present an anode-free design to balance the electrode utilization in zinc-ion hybrid supercapacitors by introducing copper nanoparticles to improve the Coulombic efficiency and cycle life of zinc anodes and activating carbon cathode by an electrolyte reaction to increase the gravimetric capacity. The following serious issues limit the publication of this submission.

1. Some key points of this work have been reported in the literature, such as the construction of activated carbon-based zinc-ion hybrid supercapacitors (e.g., doi.org/10.1016/j.ensm.2018.01.003; doi.org/10.1002/aenm.202202303), the optimization of the zinc deposition interface using Cu nanoparticles (e.g., doi.org/10.1002/adma.202300019; doi.org/10.1002/adma.202203835; doi.org/10.1002/ange.202212587), and the design of anode-free zinc-ion batteries (e.g., doi.org/10.1038/s41467-022-35630-6; doi.org/10.1021/jacs.1c12764). The authors should not miss them since these representative reports are the basis of this work.

Reply: Among these seven suggested papers, four of them are already cited in our original manuscript, as Ref. 17 (doi.org/10.1002/adma.202203835), Ref. 33 (doi.org/10.1002/ange.202212587), Ref. 7 (doi.org/10.1038/s41467-022-35630-6) and Ref. 12 (doi.org/10.1021/jacs.1c12764). The other three are newly added as Ref. 35

(doi.org/10.1002/adma.202300019), Ref. 36 (doi.org/10.1016/j.ensm.2018.01.003), and Ref. 39 (doi.org/10.1002/aenm.202202303).

2. Since zinc was pre-deposited on the AC cathode and then deposited on the Cu foil as the anode current collector (e.g., Fig. 2), the system cannot be called “anode-free”.

Reply: We based our use of the “anode-free” terminology on the definition that that the anode started out with a current collector only, but we understand the reviewer’s point that our approach used Zn plated from the cathode. We dropped the “anode-free” description from our title, but we still included an explanation in the introduction to clarify our difference with other papers using Zn foils as anodes.

3. Zinc deposition corresponds to a smaller nucleation overpotential, which is favorable for fast zinc nucleation. Why is this “energetically favorable for zinc to ... grow laterally on CuNPs across the surface”? In other words, why does a smaller nucleation overpotential result in a lateral deposition of zinc? Moreover, the reason that CuNPs result in Zn(002) deposition and high Coulombic efficiency is not well revealed.

Reply: Our results are in agreement with a prior work (Ref 37: Yan et al. ACS Nano 2022, 15, 915) in which Cu(100) substrate was shown to preferentially favor Zn(002) growth. The explanation in this reference was that the Cu(100) lattice offered a well matched lattice for Zn(002) deposition, and their data showed a decreased overpotential for zinc plating over Cu(100) compared to a routine Cu foil. While our RF sputtered Cu nanoparticles are not the same as the Cu(100) foil, there is the similarity in that a smaller nucleation overpotential is correlated to more prominent Zn(002) orientation, as shown in the figure below. We modified our explanation to “The decrease in nucleation overpotential was in similarly observed in Ref. 37, which was correlated to preferential Zn(002) deposition over Cu(100) compared to a routine Cu foil.” As for the Coulombic efficiency, we added a clarification on p.6 that “the Cu nanoparticles and bare Cu surfaces actually showed similar Coulombic efficiency values, but the difference is the extended cycle life for the Cu nanoparticles surface due to dendrite suppression.”

Comparison of (A) our overpotential data in Figure 3D to (B) a figure of overpotential from Ref. 37, showing that a decrease in overpotential is correlated to more preferential Zn(002) deposition.

4. The test method of Coulombic efficiency in Fig. 3G should be described in detail. Besides, it seems that for the Coulombic efficiency points in Fig. 3G, some are 99.9% and some are only about 99.6%. How can the average Coulombic efficiency reach 99.9%?

Reply: We added the following clarifications to p.12: “The Coulombic efficiency was measured with a half-cell setup with Zn foils serving as the counter and the reference electrodes. To determine the Coulombic efficiency of the electrode, zinc was deposited to the capacity of 0.45 mAh cm⁻² on the current collector under different current densities, and then it was stripped away to a cutoff potential of 0.5 V vs. Zn/Zn²⁺ at the same current density. And, the coulombic efficiency of the device = (stripping capacity)/(deposition capacity), with the capacities calculated from the GCD characteristics.

To clarify Fig. 3G, we modified the description on p.6: “In Figure 3g, the Coulombic efficiency (C.E.) was 99.6% (squares, at 26 mA cm⁻²) to 99.9% (circles, at 130 mA cm⁻²) for the anode with CuNPs after 1000 cycles, while the anode with a bare Cu surface short-circuited after just 200 cycles, irrespective of the applied current densities being high or low. While the CuNPs and bare Cu surfaces showed similar C.E., the difference was the extended cycle life for the CuNPs surface due to dendrite suppression.” We want to point out that the two efficiency values were taken at two different current densities, and we did not average them.

5. Rate performance of the Cl₂-activated carbon cathode should be evaluated further at large current densities such as 5-20 A/g (Fig. 4E). The current density of 0.8 to 80 mA/g is too small, which cannot reflect the performance of the carbon cathode for supercapacitors. Besides, N₂ or CO₂ adsorption-desorption tests are needed to study the pore structure of the Cl₂-activated carbon material.

Reply: Thank you for the correction. The current unit “mA g⁻¹” in Figure 4E is wrong. It should be 0.8- 80 A g⁻¹. In our previous draft, the weight should have been stated in mg (mA mg⁻¹), not in g, and that’s why we were off by that factor. We corrected Fig. 4E and Supplemental Fig. S11.

And as mentioned earlier in our reply to Reviewer1 comment#2, we added the BET measurements via N₂ adsorption-desorption in Supplemental Figure S11 to show the increase in surface area in the Cl₂-activated materials. We added the explanation on p.6 of the main text that “The active surface area of the AC on a carbon cloth substrate was measured to be 1670 m² g⁻¹ for the as-purchased AC, in good agreement with the specification of 1600 m²/g provided by the vendor Kuraray. Then the surface area of this sample increased by 3.4 times to 5750 m² g⁻¹ after the Cl₂ activation. The half-pore width also expanded from sub-nanometer to 1.5 nm after Cl₂ activation, confirming the larger pores seen in the SEM images.”

6. Since carbon fiber cloth was used as the current collector of the carbon cathode, its effect on the calculated capacity of the AC cathode should not be ignored. Especially the carbon fiber cloth could also be activated by the Cl₂ which provided a considerable charge-storage capacity, making the measured capacity of the AC cathode notably enhanced.

Reply: Thank you for this comment, and this is also raised by Reviewer1 comment#1. As we stated in our earlier reply, the specific capacity of Cl₂-activated carbon cloth (current collector) is

only 0.147 mAh g^{-1} . In the cathode (mass of carbon cloth: 9.7 mg ; mass of AC: 1.61 mg), the capacity contribution from the Cl_2 -activated current collector is only 0.48% compared to the whole cathode. The clarification is added to p.6 and as Supplemental Figure S11.

7. The charge/discharge profiles at different current densities of the “anode-free Zn-AC supercapacitors” should be provided to check the Coulombic efficiency and rate capability of the devices.

Reply: The charge/discharge profiles of the full supercapacitors at different temperatures were added as Supplemental Figure S17. At -60°C , the rate capability was tested at a lower range due to the slower kinetics and a higher ohmic resistance compared to higher temperatures.

Added in Supplemental Figure S17. Voltage-capacity profiles of the supercapacitor in Figure 5E at various charge/discharge currents and at different temperatures.

We thank the reviewers and the editor for your time and consideration. Our point-by-point response is below.

Reviewer #1 (Remarks to the Author):

The authors have made a thorough revision to their paper, which should meet the high standard of Nat. Commun. I think this manuscript can be accepted without further revision.

Response: Thank you very much for the positive review!

Reviewer #2 (Remarks to the Author):

After carefully checking the authors' responses, I still do not recommend the publication of the submission.

(1) I pointed out similar works in the aspects of the construction of activated carbon-based zinc-ion hybrid supercapacitors, the optimization of the zinc deposition interface using nanoparticles and the design of anode-free zinc-ion batteries, but this work does not present enough novelty compared with these literature.

Response: We respectfully disagree with this statement, because the prior work mentioned by the reviewer did not present the following novel aspects in our work:

(1) Regarding the construction of activated carbon-zinc hybrid supercapacitors: We introduce a novel Cl_2 activation method to enhance the porosity of activated carbon (AC). Based on our literature survey, this is a unique cathode treatment strategy. Our method increases the capacity of Cl_2 -activated AC by nearly three-fold compared to untreated baseline materials as purchased from vendors, enabling the device to demonstrate capacity comparable to or even exceeding that of reported Zn-ion batteries. Moreover, as discussed later in our response to Comment 4, the larger pore sizes created during the Cl_2 activation process further enhances ion transport and charge storage to allow state-of-the-art high-power output in the hybrid supercapacitors. As Reviewer#2 raised a concern on the device kinetics in Comment 4, it further underscores why achieving such high power and fast kinetics is novel.

(2) Regarding the optimization of zinc deposition interface using nanoparticles: The papers mentioned by Reviewer#2 include studies that use C/Cu nanocomposite layers, antimony/antimony-zinc alloys, CuZn alloys, etc. as interface modification strategies. However, these works did not systematically vary the current density to induce a strong (002) crystal facet preference. Our work studied this effect, and our results show that high current density increased preferential deposition on the (002) facet. This finding offers a new promising outlook for extending the cycle life of Zn ion supercapacitors, since they are often operated at high current densities.

(3) Regarding the anode-free design: We used the “anode-free” definition from references including Yang, et al. Adv. Func. Mater., DOI:10.1002/adfm.202400839; Alshareef et al., JACS, DOI: 10.1021/jacs.1c12764. We are inspired by their anode-free battery design, where the Zn

source is derived from the cathode storage and deposited on the anode on the first charge process. Thus, initially there is only a current collector foil on the anode side, with the active Zn stored on the cathode side. This design eliminates the need for thick Zn foil (e.g., 100 μm) as the anode, thereby reducing both the mass and volume of the device. For hybrid supercapacitors, unlike battery cathodes, the AC cathode would not contain any active Zn, so some Zn must be pre-deposited on the cathode, which we have explicitly stated in Figure 2 of the main text. With this design, we eliminated the need of Zn foil as the anode, in contrast to prior work on Zn ion supercapacitors. Our design worked towards increasing Zn utilization and balancing n/p ratio down to 2.5, closer to the ideal ratio of 1 with better performance than previous studies of metal ion supercapacitors.

(2) The concept of "anode-free" is not suitable for this work. Anode loadings are listed in Table 1, which are not zero.

As discussed in our response to Comment 1, our anodes started out as current collectors only, with zero Zn loading, as shown in Figure 2 of the main text. The anode loadings in Table 1 refers to the active Zn that was stored on the cathode side, which was then deposited onto the anode current collector in the first charging cycle. Thus, the device did start out without any Zn on the anode in the first cycle. However, as our n/p ratio was at least 2.5 and not at the ideal value of 1, there was still excess Zn compared to AC capacity. We acknowledged this caveat and removed "anode-free" from the manuscript; instead, we modified a sentence on page 7 to highlight our goal of balancing electrode utilization: "The supercapacitors were prepared at a n/p ratio of 4 and 2.5 (n: Zn; p: AC), to demonstrate high utilization of active materials."

(3) The question that why the Cu nanoparticles are favorable for zinc to grow laterally to form (002) horizontal platelets is not well answered.

The electro-crystallization orientation of Zn on Cu foils is influenced by two key factors: Cu nanoparticles (CuNPs) and high current density. The measurement evidence in Fig. 3 and Supplemental Figs. S3-S7 clearly demonstrated the preference for (002) growth. Regarding the role of CuNPs, we presented evidence in Fig. 3d and Supplemental Fig. S5 that the plating overpotential of Zn on CuNPs was smaller than on bare Cu, which we attributed as one explanation that the energy barrier for Zn nucleation was reduced by CuNPs, allowing for more uniform lateral (002) plating (citing Refs. 37 and 38: Energy Environ Sci. 2024, 17, 369 and ACS Nano 2022, 16, 9150, respectively). Complementary to this energy view, another work (Ref. 39: Nanomicro Lett, 2022, 14, 39) presented finite element analysis (FEA) simulations that showed a more uniform electric field distribution due to nanowire or 3D scaffolds. This electric field effect might be applicable to CuNPs, since they would likely change the surface electric field in a similar manner. Another consideration is the binding energy of Zn on different Cu surfaces. From density functional theory (DFT) calculations, the edges of Cu nanowires would reduce Zn binding energy (Ref. 39). A similar mechanism might apply to CuNPs due to the

increase in edges, promoting zincophilicity and dense lateral plating. We added these explanations on p. 5 in the main manuscript, referencing the FEA and DFT work. We believe that a more detailed exploration of these mechanisms would be beyond the scope and focus of this paper and should be allocated to future work.

(4) The authors directly changed the current density from 0.8-80 mA/g in the previous manuscript to 0.8-80 A/g in the current manuscript. How can the water-in-salt electrolyte (15 M ZnCl₂) and the highly porous carbon cathode realize such superior fast-rate performance?

As explained in our previous response, the labeling of "mA/g" in Figure 4E was a typo, and we apologize for the mistake. Other sets of galvanostatic charge-discharge (GCD) data with fast-rate performance are also shown in Supplemental Figure S17, and to which we added another set of cyclic voltammetry (CV) data copied below to provide more evidence that the reported current density is accurate. At a scan rate of 10 mV/s, the measured peak current was ~12 mA, for active materials loading of 1.72 mg. Thus, the current density is 12 mA/1.72 mg = 7 A/g for the CV curve, showing performance in agreement with the range of values in Figure 4E. For transparency, all data have been deposited in the data sharing website (figshare <https://doi.org/10.6084/m9.figshare.28040456>) for public access.

(Left plot) added to Supplemental Figure S17: CV curves with scan rates of 1 mV/s to 10 mV/s for the zinc ion supercapacitor. (Right plot) added to Supplemental Figure S11: Imaginary impedance versus real impedance of AC before and after Cl₂ activation, measured at 1 V versus a Zn/Zn²⁺ reference.

The reviewer's comment asks why the device can operate at high rate with water-in-salt (WIS) electrolytes and activated carbon electrode. Regarding the electrolyte's role, while WIS are often thought to be viscous slowing down ionic conductivity relative to aqueous systems, they still show high ionic conductivity (0.024 S/cm at 25°C for 15 M WIS and 0.18 S/cm at 25°C for 7.5 M WIS as shown in Supplemental Figure S16). This level of ionic conductivity is similar to organic electrolytes used in other supercapacitors with high-power output (Ref 11 in the main text), and therefore the ionic conductivity of WIS is not the limiting factor to rate performance.

Rather, the diffusion resistance in the electrode bulk is more limiting, and the reviewer is right to be concerned about kinetics due to AC cathode. However, for our cathodes, one of the key

novelties of this work is the Cl_2 activation treatment applied to the porous cathode. This treatment significantly increases the porosity of the AC and enlarges the pore size (Supplemental Fig S10), facilitating easier ion diffusion within the AC structure. As a result, the electrochemical kinetics of our activated AC are much improved compared to commercial porous carbon materials. This improvement is further supported by our electrochemical impedance spectroscopy analysis, as shown in Supplemental Fig. S11 and copied above. The cathode resistance is reduced by 1.2 ohm after Cl_2 activation to enable the fast-rate performance.